# Integration Vis-NIR Spectroscopy and Artificial Intelligence to Predict Some Soil Parameters in Arid Region: A Case Study of Wadi Elkobaneyya, South Egypt

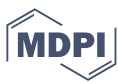

**Moatez A. El-Sayed [1], Alaa H. Abd-Elazem [2], Ali R. A. Moursy [3], Elsayed Said Mohamed [4,5,*], Dmitry E. Kucher [5] and Mohamed E. Fadl [6,*]**

1. Soils and Water Department, Faculty of Agriculture, Al-Azhar University, Assiut 71524, Egypt
2. Soil and Natural Resources Department, Faculty of Agriculture and Natural Resources, Aswan University, Aswan 81528, Egypt
3. Soils and Water Department, Faculty of Agriculture, Sohag University, Sohag 82524, Egypt
4. National Authority for Remote Sensing and Space Sciences, Cairo 11843, Egypt
5. Department of Environmental Management, Institute of Environmental Engineering, People's Friendship University of Russia (RUDN University), 6 Miklukho-Maklaya Street, 117198 Moscow, Russia
6. Division of Scientific Training and Continuous Studies, National Authority for Remote Sensing and Space Sciences (NARSS), Cairo 11769, Egypt
* Correspondence: salama55@mail.ru (E.S.M.); adhamnarss@yahoo.com (M.E.F.)

**Abstract:** Understanding and determining soil properties is reflected in improving farm management and crop production. Soil salinity, pH and calcium carbonate are among the factors affecting the soil's physical and chemical properties. Hence, their estimation is very important for agricultural management, especially in arid regions (Wadi Elkobaneyya valley, located in the northwest of Aswan Governorate, Upper Egypt). The study objectives were to characterize and develop prediction models for soil salinity, pH and calcium carbonate ($CaCO_3$) using integration soil analysis and spectral reflectance vis-NIR spectroscopy. To achieve the study objectives, three multivariate regression models: Partial Least Squares Regression (PLSR), Multivariate Adaptive Regression Splines (MARS) and Least Square-Support Vector Regression (LS-SVR)); and two machine learning algorithms, Random Forest (RF) and Artificial Neural Networks (ANN) were used. Ninety-six surface soil samples were collected from the study area at depths 0–5 cm. The data were divided into a calibration dataset (70% of the total) and a validation dataset (30% of the total dataset). The obtained results represent that the PLSR model was the best model for soil pH parameters where $R^2$ of calibration and validation predictability = 0.68 and 0.52, respectively. The LS-SVR model was the best model to predict soil Electrical Conductivity (EC) and soil Calcium Carbonate ($CaCO_3$) content, with $R^2$ 0.70 and 0.74 for calibration and $R^2$ 0.26 and 0.47 for validation, respectively. On the other hand, the results of the implemented machine learning algorithm model showed that RF was the best model to predict soil pH and $CaCO_3$, as the $R^2$ was 0.82 for calibration and 0.57 for validation, respectively. Nevertheless, the best model for predicting soil EC was ANN, with an $R^2$ of 0.96 for calibration and 64 for validation. The results show the advantages of machine learning models for predicting soil EC, pH and $CaCO_3$ by Vis-NIR spectroscopy. Therefore, Vis-NIR spectroscopy is considered faster and more cost-efficient and can be further used in environmental monitoring and precision farming.

**Keywords:** soil parameters; vis-NIR; statistical parameters; remote sensing; Wadi Elkobaneyya

## 1. Introduction

Soil is a complex system and is considered a primary natural resource for food production, energy, and the food chain. Egypt covers an area of almost one million square kilometers in North Africa and Western Asia; the Nile Valley has a flood plain of about 18 km wide, bordered by flat terraces. The Delta, however, has an area of 220 km wide at

the coastline and is 170 km long [1]. Soil is affected by surrounding factors, such as topography, climate factors and etc. The change in soil properties is reflected in its quality [2,3], hydrological characteristics [4], fertility status and a probable carbon sink to alleviate global warming phenomena [5–9]. For that purpose, the soil survey is the main source of information to assess soil suitability [10,11]. In practice, soil samples represent a composite material of 15 to 20 soil samples to cover an area of 12 to 20 hectares of the study's areas [12]. To match human needs with the limited land resources, a larger amount of spatial soil data is needed as a step forward for precision agriculture [13,14]. The most authoritative method for soil analysis is the traditional or conventional laboratory method, but it's very costly and time-consuming for soil sample collection and analysis for a specific purpose [15]. In addition, it needs a lot of preparation stages and also large amounts of chemical materials for the determination of the soil properties [16,17]. Therefore, there is an urgent need for new methods of soil analysis which are faster and more cost-efficient [18,19].

Therefore, reliance on advanced methods for predicting soil characteristics is becoming required for environmental monitoring and precision agriculture. A diffuse reflection spectroscopy (DRS) method that is directly affected by the absorption properties of the incident electromagnetic spectrum [20–22]. Imaging Spectroscopy (IS) has proven to be a vital tool for spatially distributing soil properties and generating maps. Moreover, an airborne-based Hyperspectral Remote Sensing (HSRS) image includes the vis-NIR and Short-Wave Infrared (SWIR) regions which offer the possibility of mapping soil properties [23]. Imaging spectroscopy was used for studying and mapping the spatially distributed soil characteristics, such as soil EC using the integration of DRS and imaging sensors [24]. Over the past 35 years, soil spectroscopy provided a promising capability for identifying vegetation, rocks, and minerals [25]. DRS is a modern technology that also has been proven to be highly efficient for estimating soil parameters. It is faster and cheaper than conventional methods. Also, these tools are environment-friendly, non-destructive, reproducible, and repeatable in analytical methods. The DRS technique is applied in both field and laboratory conditions to calculate several soil characteristics without soil sample preparation [16]. The spectral reflectance that ranged between 0.35 and 2.5 μm is more suitable for estimating the majority of soil parameters [26]. Nowadays, quantitative soil parameter prediction using spectroscopy, multivariate statistics and chemometrics techniques is still growing, and the possibility of soil property estimation increased after new high Signal-to-Noise Ratio (SNR), and hyperspectral sensor availability increased, which can be used in the field, laboratory, or fixed on airborne platforms [27]. Spectra-based remote sensing is used in several fields, e.g., space-borne and laboratories [26]. Recently, these new techniques have become more rapid and accurate in soil characteristic estimation [28]. Several studies used spectra-based remote sensing techniques for determining soil parameters, such as texture, clay mineralogy and soil $CaCO_3$, and these techniques are more significant and cost-effective for large-scale soil parameter predictability [29–31]. O'Rourke and Holden (2011) [32] showed good results of integrated laboratory-based hyperspectral sensing and vis-NIR spectroscopy for Soil Organic Carbon (SOC). The authors reported that this method was ten times cheaper than the conventional methods. The spectroradiometer and hyperspectral sensor reflectance techniques were able to characterize the soil based on the spectral soil data collected. Hence, Soil Spectral Libraries (SSL) were generated in many regions after covering the majority of soil variations using many processes for data analysis qualitative, such as discriminant analysis, bands ratio and image classifications [33,34]. DRS technique has proven to be a highly efficient, environment-friendly, non-destructive, reproducible and repeatable analytical method for estimating soil characteristics [35]. Many studies applied DRS for soil pH estimation, whereas the spectra are highly sensitive to carbon. EC is considered an indirect indicator of soil's physical properties. The spectral reflectance is affected in wavelengths of around 1400 and 1900 nm. Similarly, the spectra are also affected strongly in saline soils compared to moderate saline soils due to stronger water absorption features. Most sensitive spectra have been recorded as 390, 615, 685, 800, 950, 1410,

1935, and 2350 nm. The organic matter contents of >2% affected the absorption of soil spectral features. The visible region is the most sensitive and responds to the organic matter [36].

The integration of machine learning and DRS for the accurate prediction of soil parameters has become a promising tool for saving time, cost, and effort. Multivariate algorithms are commonly used to model soil parameters based on soil spectra, such as Partial Least Square Regression (PLSR), Artificial Neural Networks (ANN), Multivariate Adaptive Regression Splines (MARS), and Random Forest (RF) [37]. The MARS model produces a non-parametric regression model and a generalization of recursive division regression approaches, which generates multi-definition linear models (price-wise) instead of multi-definition static models [38]. Random Forest is a promising approach for soil parameter prediction in which samples are drawn to construct multiple trees, but the difference is that each tree is grown with a randomized subset of predictors [39]. Root Mean Square Error (RMSE) and Ratio of Performance Deviation (RPD) were reported as prediction errors for soil parameter estimations and validation of that concentration from spectral data. The coefficients of multiple determinants as the correlation square ($R^2$) between response and predicted values were also computed as unsuitable regression [40]. Ramdas et al. (2015) [41] recommended that Principal Component Regression (PCR) and PLSR algorithms are widely used to extract soil properties data during vis-NIR to Mid. Infrared (MIR) spectra range. Mapping based on implemented the spectral models on airborne images has better assessment and accuracy for soil parameters [42]. Mustard and Sunshine (2003) [43] used the NIR spectra and PCR method for predicting soil minerals (Kaolinite and Montmorillonite), clay and soil characteristics, such as Cation Exchange Capacity (CEC), SOC, and extractable iron (Fe). Brown et al. (2006) [44] found that Boosted Regression Trees (BRT) model is better than Partial Least Square (PLS) in estimating clay, SOC, and CEC. Kumar et al. (2009) [45] highlighted the efficiency of the Spectroscopy data for estimating soil properties of Punjab using a stepwise regression approach with high accuracy (i.e., $R^2 = 0.93$ for $CaCO_3$ and 0.68 for Nitrogen (N)).

LS-SVR was used for the classification and regression analysis of linear and nonlinear multivariate problems, using linear equations set and not quadratic programming. It has been widely used in the sector of chemometrics, such as in soil spectroscopy which is a highly nonlinear [46], therefore; a normal SVR that is usually utilized for linear classification can result in poor prediction capability. Hence, it needs to be expanded for nonlinear regression by using a kernel function [47,48].

Nowadays, the majority of soil researchers are using RS, particularly reflectance spectroscopy and GIS, for soil mapping because these techniques are cheaper and cost-effective [23,49,50]. Many researchers applied multivariate regression models integrated with vis-NIR to achieve an accurate quantitative estimation of soil parameters [51]. Soil maps are the source of information for a better understanding of soil properties and land management, whereas; the conventional techniques of soil mapping are expensive and consume time [7].

The research objectives are as follows:

1. To characterize soils using hyperspectral reflectance data collected from the ground sensor.

2. To develop prediction models for Soil EC, pH and $CaCO_3$ using reflectance vis-NIR Spectroscopy.

## 2. Materials and Methods

### 2.1. Description of the Study Area

Wadi Elkobaneyya valley is a part of the Western Desert, about 20 km from the northwest part of Aswan city. Its lies between $32°45'8.788''$–$32°53'00''$ N and $24°12'18.546''$–$24°19'7.458''$ E, as represented in Figure 1, and this study area covers about 42.34 km$^2$ in the Aswan Governorate, Egypt.

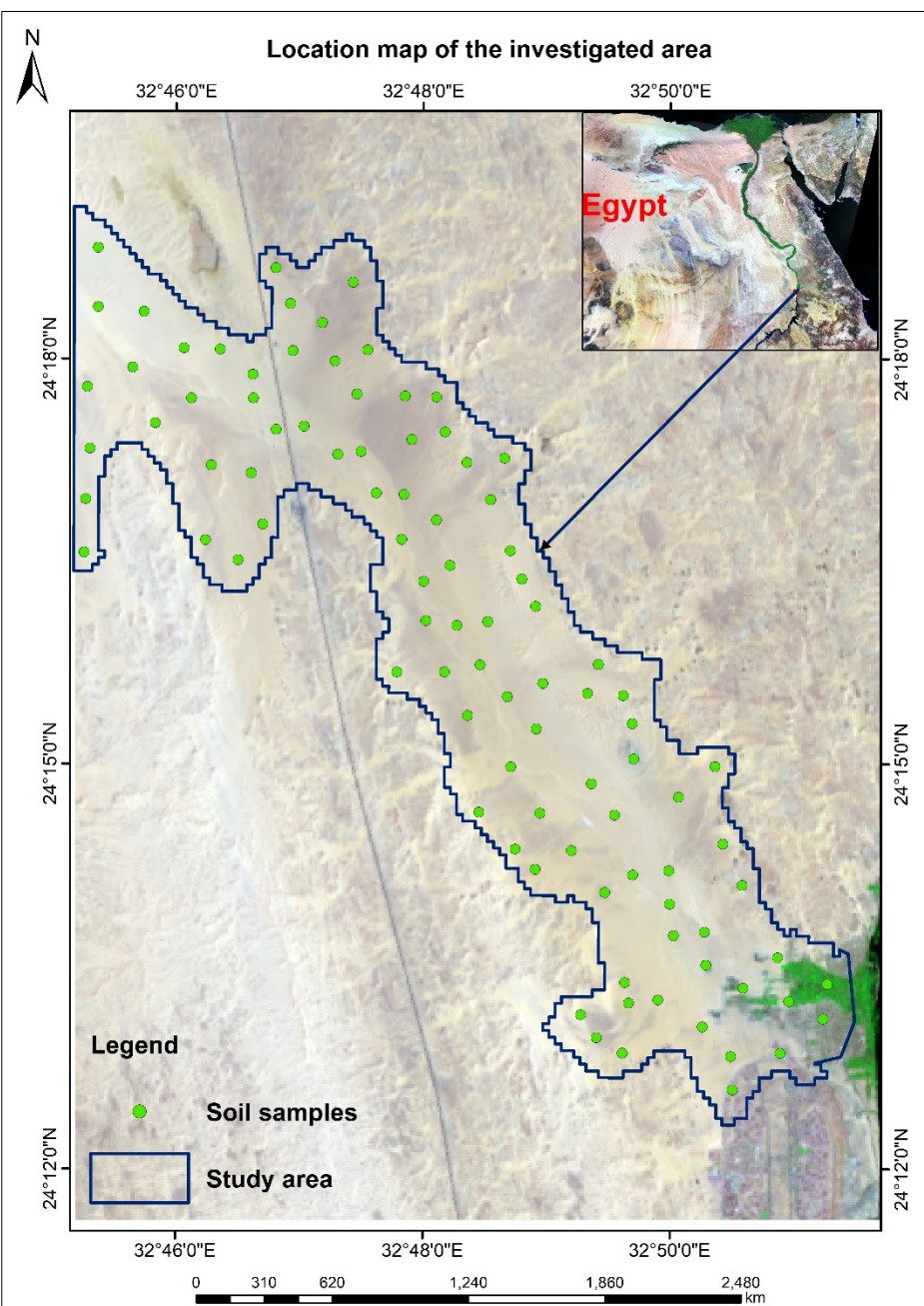

**Figure 1.** Location maps of the investigated area (Landsat-8, 2022).

The study area is characterized by hot and dry summers with little rainfall in winter and bright sunshine throughout the year. Where the surface temperature is between 22.9 °C in winter and 41 °C in the summer period. The mean annual precipitation is about 0.85 mm, as shown in Table 1. Soils in Elkobaneyya valley are generally characterized by a Hyperthermic soil temperature regime and Torric soil moisture regime, soils are mainly calcareous, and the common soil orders are Aridisols and Entisols according to the United State Department of Agriculture (USDA) soil taxonomies [52]. The elevation of the investigation area varies from 78 to 196 m Above Sea Level (ASL). The Nubian sandstones are the most important sediment rocks covering the study area. Generally, Quaternary sediments occupy most of the studied area. They were represented by Aeolian sand, sand accumulations and salt crusts [53].

**Table 1.** Meteorological data of the investigated area.

| Climate Data for the Study Area | | | | |
|---|---|---|---|---|
| Month | Minimum | Maximum | Average | ST.DEV. * |
| Record high Tem. * °C | 35.30 | 51.00 | 45.10 | 5.13 |
| Average high Tem. * °C | 22.90 | 41.40 | 33.62 | 6.75 |
| Daily mean Tem. * °C | 15.30 | 33.60 | 26.01 | 6.79 |
| Average low Tem. * °C | 8.70 | 26.00 | 18.48 | 6.28 |
| Record low Tem. * °C | 0.60 | 20.00 | 9.90 | 7.19 |
| Average rainfall mm | 0.00 | 1.40 | 0.22 | 0.43 |
| Average rainy days (≥0.01 mm) | 0.00 | 0.85 | 0.13 | 0.26 |
| Average relative humidity (%) | 16.00 | 42.00 | 26.17 | 8.79 |

Source: NOAA, 2022 for mean temperatures, rainfall, humidity, Meteo. Climate. Tem. *; Temperature. St. Dev. *; Standard Deviation.

## 2.2. Soil Sampling and Analysis

The fieldwork was conducted in February 2022, and soil samples were geo-referenced using the Global Positioning System (GPS), as shown in Figure 1. The soil samples were dug and described according to the standard scheme and terminology of the Food and Agriculture Organization (FAO, 2006) [54] and American Soil Survey Staff (2014) [52]. A total of 96 soil samples were collected and covered the recognized different soil layers in Table 2.

**Table 2.** Soil properties descriptive statistics data.

| Soil Properties | | | |
|---|---|---|---|
| | $CaCo_3$% | pH 1:2.5 | EC (dS/cm) |
| Mean | 1.60 | 7.83 | 0.70 |
| Standard Deviation | 1.92 | 0.73 | 0.40 |
| Sample Variance | 3.68 | 0.53 | 0.16 |
| Minimum | 0.04 | 5.11 | 0.22 |
| Maximum | 9.40 | 8.67 | 2.65 |
| Count. | | 96.00 | |

Soil Chemical Properties

a.  Total calcium carbonate ($CaCO_3$) was determined using Scheibler's calcimeter [55].
b.  Soil reaction (pH) was measured at 25 °C using a glass electrode according to Alvarenga et al. (2012) [56].
c.  Soil salinity was determined as EC in soil extract using the Beckman conductivity bridge at 25 °C according to Bashour and Sayegh (2007) [57].

## 2.3. Processing and Analysis of Soil Spectral Data

### 2.3.1. Ground-Based Spectral Data Pre-Treatment

Soil spectral data collected using the ASD spectroradiometer in the laboratory conditions were arranged in text format (CSV files) for the processing stage. The obtained spectral data were in 1 nm intervals and converted to be in 5 nm intervals.

### 2.3.2. Models Development and Statistical Analysis

Prior to the modeling process, preliminary processing of the soil spectral readings was performed using a multiplicative scatter correction (MSC) [58] to choose an appropriate specific wavelet function and scale.

Multivariate regression models are used to model soil parameters. Many regression models are applied in soil studies, such as PLSR, ANN, MARS, ANN and RF. In the modeling part, randomly selected 70% of the soil samples were selected for calibration models for different soil properties. Rest data records (30%) were used for model validation.

Partial Least-Squares Regression (PLSR)

The PLSR is used as a developed prediction model for quantitative spectral analysis based on highly collinear predictor variables. The PLSR algorithm is running to select the orthogonal factors that increase the predictor (X spectra that are the mean-centered before decomposition) and response variables (lab data) variance. PLSR disintegrates X and y into factor scores (T) and factor loading (P and q). The remaining noise factors can be ignored. Hence residues E and f are added, as represented in the next Equations (1) and (2) [59].

$$X = TP+E \tag{1}$$

$$y = Tq+f \tag{2}$$

The calibration models and RMSE error of predictions were computed to select the optimal leave-one-out cross-validated calibration model and predict each soil parameter individually [60].

The R studio software 4.1.2 PLS package was running to develop the different soil parameters calibration and validation models using soil vis-NIR spectral data and the laboratory soil data (96 soil samples were randomly split into two groups, 70% of the data for calibration model and 30% used for setting the validation model) through the following stages [61]:

(i)    Data normalization (0 and 1 values);
(ii)   Data dividing (into two data sets; 2/3 for the calibration data set and 1/3 for the validation data);
(iii)  Data sorting (depending on their weights among the calibration and validation data sets); and
(iv)   Data outliers' removal (remove the much higher or lower soil parameter values using a suitable method).

The Box–Cox method was used to remove outliers according to (Box and Cox 1964) [62]. The function of 'inv-BoxCox' was used in the (R) studio software, whereas; it applied to all calibration and validation datasets. Box–Cox transformation is a statistical technique that involves transforming the target variable (soil parameter) so that the data is subject to a normal distribution. The Box–Cox transformation helps to improve the predictive power of the calibration and validation models because it removes the noise (the outliers), Equation (3).

$$w_t = \begin{cases} \log(y_t) & \text{if } \lambda = 0; \\ (y_t^{\lambda}-1)/\lambda & \text{otherwise.} \end{cases} \tag{3}$$

where: t is the time period (not included because the data is non-time series) and $\lambda$ is the parameter that was chosen; w is the transformed data of the targeted soil parameter y.

Multivariate Adaptive Regression Splines (MARS)

MARS is a non-parametric regression model introduced by Friedman (1991) [63]. It basically determines the relationship between the predictor's dependent and set variables by fitting multiple definition linear regressions according to flexible models building [64]. The general MARS model equation is represented in the next Equation (4).

$$y_p = \alpha_0 + \sum_{m=0}^{M} \alpha m BF_m(x) \tag{4}$$

where: $y_p$ is the predicted dependent variable; $M$ is the number of $BF_m$ data; $\alpha_0$ is the constant term; $\alpha m$ is the coefficient of single spline function; and $m$ and $BF_m(x)$ are the m[th] truncated spline functions.

Least Square-Support Vector Regression (LS-SVR)

The LS-SVR model is used to predict the values of each soil parameter which can be used to find the best-fit line of the entire dataset; the best-fitted line of the dataset includes the maximum number of points fitted with the targeted variable (soil parameter). However,

the SVR model was used in modeling the predictability of each soil parameter in this study (soil pH, soil EC and soil CaCO$_3$).

In the current study, an LS-SVR is used with the Gaussian Radial Basis Function (RBF) kernel as a training algorithm with polynomial kernels (Equation (6)). The RBF kernel algorithm requires two parameters for tuning, namely gamma (Ɣ), which is the regularization parameter that determines the trade-off between the training error minimization and smoothness [65], and σ² which is the squared bandwidth of the Gaussian curve. For the tuning of these parameters, leave-one-out cross-validation is used for choosing the initial random parameters [66], to be optimized by means of the standard simplex method [67].

$$K\ (X_i, X_j) = \exp\left( -\frac{\|X_i - X_j\|^2}{\sigma^2} \right) \tag{5}$$

where: Ɣ = 1/2 σ²; $K$ is a kernel radial basis function; $X_i$ and $X_j$ are vector points in any fixed dimensional space; and σ² is the squared bandwidth of the Gaussian curve.

The input parameters used for training the LS-SVR are the vis-NIR features that will be derived from the latent variables (LVs) calculated from the PLS regression model.

Random Forest (RF)

The RF classifier is a regression process of tree predictor's combination of random input vector or randomly selected variables at each node on numerical values as arbitrary to class labels [68]. The RF classifier is a process to develop a training data set by randomly drawing and using the construction of individual trees for every feature; these fully developed trees are not pruned back, and one of random forest regression's main advantages over other tree techniques [69].

The Neural Network Approach

ANN model contains the neurons minimum number of that is capable of simulating training data and feed-forward a back-propagation neural network with the Levenberg–Marquardt training algorithm has been used to find the optimal data weights [70]. Various experiments were performed using sigmoidal linear activation functions, and the over-fitting has been avoided to model development based on a number of hidden neurons selection, as shown in Figure 2. ANN equation was expressed in the following Equation (4).

$$P = f_n\left( b_0 + \sum_{k=1}^{h}\left( w_k f_n\left( b_{hk} + \sum_{i=1}^{m} w_{ik} x_i \right) \right) \right) \tag{6}$$

where: $P$ is the data prediction; $f_n$ is the transfer function; $b_0$ is the output layer bias; $h$, hidden layer neurons number; $k$ is the hidden layer neuron value; $w_k$ is the connection weight between $k$ and single output neuron; $b_{hk}$ is the bias at the $k$ and $b_0$; $m$ is the number of input variables; $i$ is the layer of input; $w_{ik}$ is the connection weight between $i$ and $k$; and $x_i$ is the input value.

The data normalization process was applied to use data sets, and RMSE and RPD were calculated to quality parameters for the accuracy assessment of the ANN model.

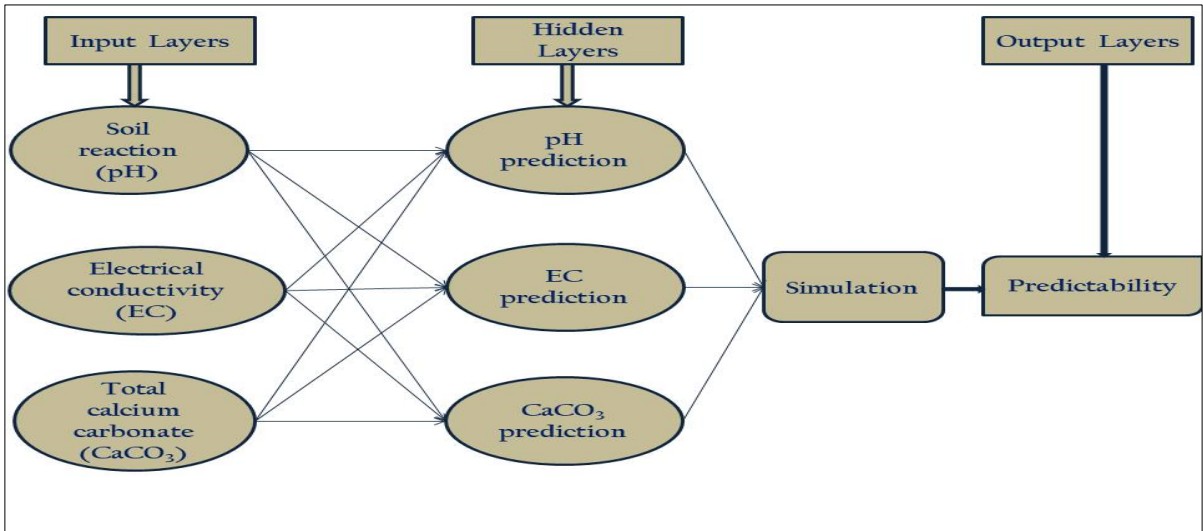

**Figure 2.** Neural Network Approach flowchart.

2.3.3. Accuracy Assessment

To validate the developed prediction models, three statistical indices were used ($R^2$, RMSE and RPD) as shown in Equations (7)–(9).

The correlation coefficient ($R^2$)

$$R^2 = 1 - \left( \frac{\sum_{i=1}^{N} (Y_i - \hat{Y}_i)^2}{\sum_{i=1}^{N} (Y_i - \hat{Y}_i)^2} \right) \tag{7}$$

where: $\hat{Y}$ represents the values estimated by the models in the $i$th observation; $Y_i$ are the values measured or observed in the laboratory in the $i$th observation; $\bar{Y}$ represents the mean of the observed values; and $N$ is the number of observations.

Room Mean Square Error (RMSE)

$$RMSE = \sqrt{1/n \sum (y - x)_2} \tag{8}$$

where: $y$ is the soil predicted values; $x$ is the soil measured value; and $n$ is the number of measured or predicted data values.

The Ratio of Performance Deviation (RPD)

$$RPD = \frac{SD}{RMSE} \tag{9}$$

where: $SD$, standard deviation.

The NIR spectra technique of different soil parameters' predictability ability is categorized into three categories according to RPD ratio and correlation coefficient ($R^2$) values; category (**A**) includes soil parameters which are highly predictable with R-square between 0.8 and 1, and with RPD above 2; category (**B**) includes soil parameters that can be predicted with a moderate performance of predictability whereas R-Square between 0.5 and 0.8, and the RPD is between 1.4 and 2 and category (**C**) that includes soil parameters which had the lowest predictability performance (R-square is lower than 0.5, and the RPD is lower than 1.4), as shown in the Table 3 [71].

**Table 3.** NIR spectra predictability categories of soil parameters.

| NIR Category | RPD | $R^2$ | Parameters |
|---|---|---|---|
| A | <2 | 1−0.8 | Moisture, sand, silt, exch. Ca, and CEC. |
| B | 2−1.4 | 0.8−0.5 | Clay, soil pH, N, K, Ca, Mg, Fe and Mn |
| C | >1.4 | >0.5 | Cu, P, Zn and Na. |

2.3.4. Variables Selection Methods

For selecting the significant bands which more related to soil chemometrics, different techniques are commonly used, such as the Competitive Adaptive Reweighted Sampling (CARS) technique that was used for vis-NIR data-sets (ASD) significant bands selecting as the optimal combination of the wavelengths existing in the full spectrum using the principal of the survival of the fittest to build a high-performance calibration model based on next steps [72]:

(1) Wavelength selection perform forced staffing;
(2) Wavelengths competitive selection realize using Adaptive Reweighted Sampling (ARS) prediction model; and
(3) Subset data evaluation based on cross-validation.

2.3.5. Competitive Adaptive Reweighted Sampling (CARS) Analysis

The CARS analysis was proposed to select the most relevant combination of variables (or wavelengths) during a successive selecting procedure. Based on the regression coefficients obtained by the PLS model, CARS iteratively selects N subsets of variables from N Monte Carlo (MC) sampling processes. During each process, fixed ratios of samples are randomly selected to establish a calibration model. Next, with the regression coefficients obtained, a two-step variable selection procedure is adopted to select the relevant wavelengths. Finally, cross-validation is used to choose the subset (the most relevant combination of wavelengths) showing the lowest root mean square error [73].

The method proceeds as follows:

Step 1: MC sampling:

Randomly select $k$ samples ($X_i$, $y_i$), i stands for the ith loop. Build a PLS model based on the dominating variables Vsel_old, then record the regression coefficients *beta* (Equation (10)).

$$beta = W * b \tag{10}$$

Step 2: Sort the variables in descending order according to the absolute value of their regression coefficients. Update the ratio of variables to be kept (Equations (11), (12) and (13)).

$$r_i = ae^{-ki} \tag{11}$$

$$a = \left(\frac{p}{2}\right)^{1/(N-1)} \tag{12}$$

$$k = \frac{\ln(p/2)}{N-1} \tag{13}$$

where; ln means the natural logarithm; and N represents the number of sampling process.

The exponent function's trace in Step 2 decreases rapidly in the first stage, whereas in the second stage, the trace progresses gently. This will facilitate the selection process [73].

Step 3: Condense the current dataset to have $p \times r_i$ variables. Then draw a subset of variables from the retained $p \times r_i$ variables using an adaptively reweighted sampling method, according to a normalized weight $w_i$ (Equation (14)).

$$w_i = \frac{|beta_i|}{\sum_{i=1}^{p} |beta_i|}, i = 1, 2, 3, \ldots, p \tag{14}$$

Essentially, the adaptively reweighted sampling method in Step 3 is a weighted sampling algorithm. The variables with larger weights will be selected with higher frequency, and this will accelerate the selection process.

Step 4: Compute RMSE using Vsel_new. Then Vsel_old = Vsel_new.

Step 5: Let i = i + 1. If i > N, return to step 1; else, continue.

Step 6: Choose the subset with the minimum RMSE as the optimal combination of variables/wavelengths and build the final calibration model.

### 2.4. Laboratory Hyperspectral Data Collection

Ninety-six ground soil samples with 2 cm thickness were scanned using Field-Spec-4 Analytical Spectral Device (ASD; Boulder, CO, USA) with wavelength ranging from 350 to 2500 nm in the laboratory condition [74]. Hyperspectral reflectance was measured under two calibrated halogen lamps (1000 W) situated at 0.70 m with a zenith angle of 30° in a dark room after sensor calibration using a white spectral panel. All recorded soil spectral signatures were converted into tab-delimited text file format using the View Spec. Pro. Software (Version 4.05) to facilitate data sharing with other software.

## 3. Result and Discussion

### 3.1. The Behavior of Soil Spectral Signatures

Figure 3 shows the spectral reflectance of soil samples; they illustrate the response and absorption areas throughout the wavelength from 1400 to 2200 nm that are associated with clay minerals, at 1400 to 1900 nm, and lattice OH (moisture adsorbed to the surface of clay) features at 1900 to 2200 nm [75].

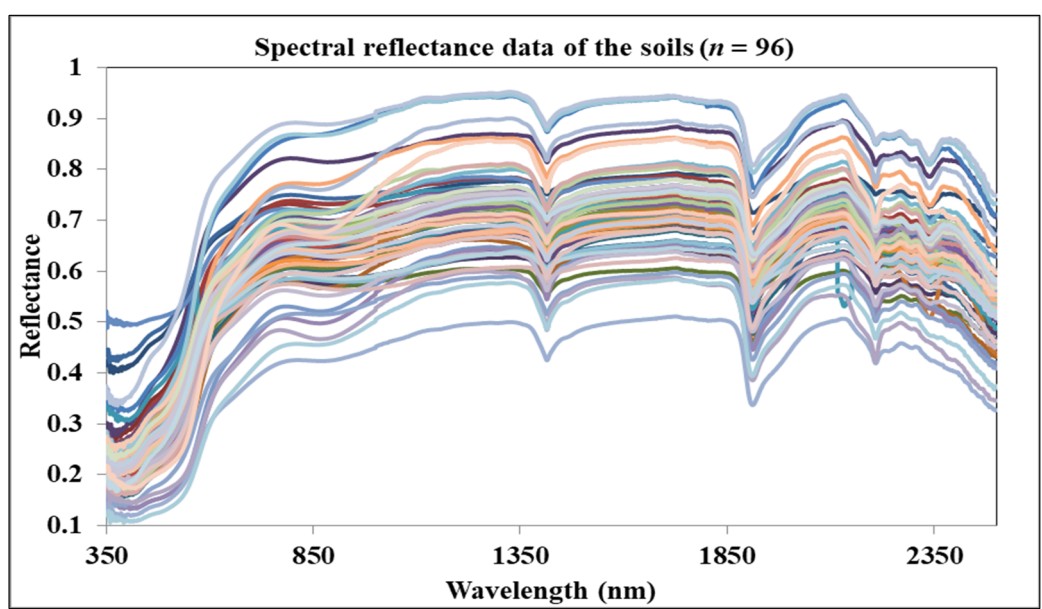

**Figure 3.** Soil samples reflectance spectra.

### 3.2. Correlation of Soil Parameters and Their Corresponding Spectral Signatures

Figure 4 and Table 4 show the correlation between examined soil parameters and soil spectral signatures, the most significantly correlated bands with each soil parameter from the obtained correlation data.

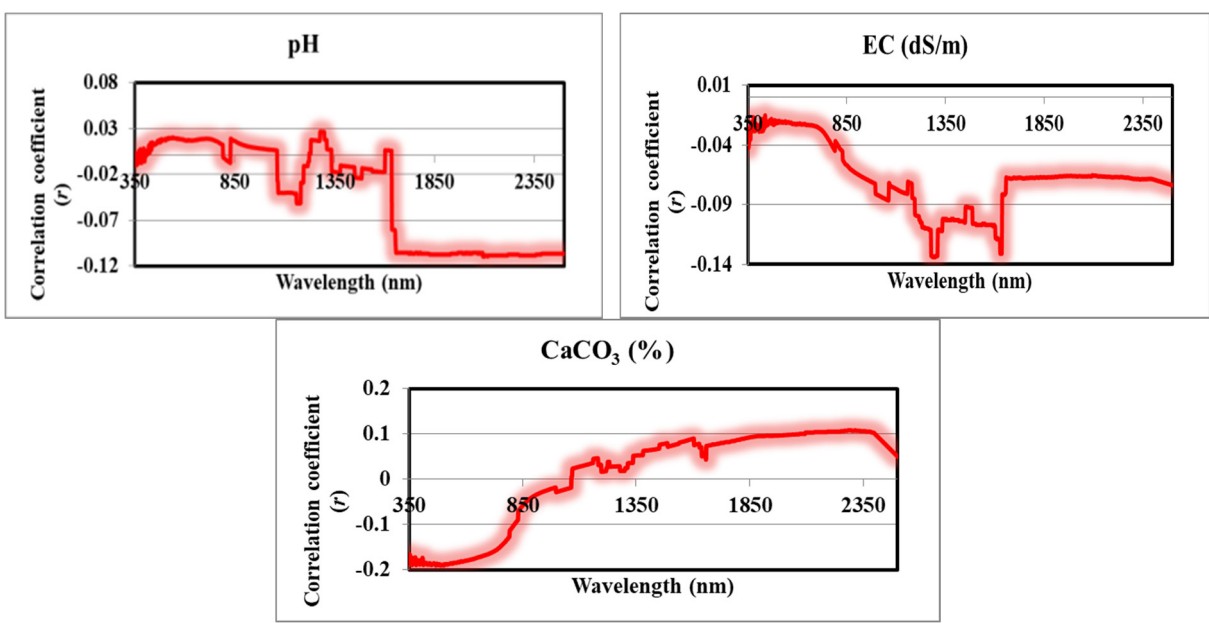

**Figure 4.** Correlation between spectra and soil parameter.

**Table 4.** The most significantly correlated bands with each soil parameter.

| pH | | | | | | | |
|---|---|---|---|---|---|---|---|
| *r* | 0.0176 | 0.0194 | 0.0271 | −0.051 | −0.0797 | −0.105 | −0.106 | −0.108 |
| Wavelengths (nm) | 492 | 828 | 1276 | 1158 | 1636 | 1656 | 2068 | 2350 |
| **EC** | | | | | | | |
| *r* | −0.0826 | −0.0985 | −0.105 | −0.133 | −0.103 | −107 | −0.118 | −0.131 |
| Wavelengths (nm) | 1014 | 1194 | 1222 | 1276 | 1410 | 1516 | 1602 | 1626 |
| **CaCO₃** | | | | | | | |
| *r* | −0.185 | −0.17 | −0.0975 | 0.0459 | 0.0679 | 0.0852 | 0.0946 | 0.107 |
| Wavelengths (nm) | 470 | 658 | 812 | 1158 | 1440 | 1564 | 1860 | 2262 |

The results were observed that in the pH soil parameter, the maximum correlation coefficient values were recorded as 0.0176, 0.0194, 0.0271, −0.051, −0.0797, −0.105, and −0.108 in wavelengths of 492, 828, 1276, 1158, 1636, 1656, 2068, and 2350, respectively.

The results agree with many research studies, such as Abdul Munnaf et al. (2019) [21,22,76], who found that the wavelengths 455, 772, 1361, and 1424 nm have sensitive responses for the concentration of pH. Like with the results of Mousavi et al. (2021) [77], the bands that responded to pH were 400–439, 499–566, 695–744, 874–883 and 885–914.

The result showed the most significant reflectance responses to (EC) were the following wavelengths: 1014, 1194, 1222, 1276, 1410, 1516, 1602 and 1626, and their correlation ®were: −0.0826, −0.0985, −0.105, −0.133, −0.103, −0.107, −0.106 and −0.108, respectively, this results corresponded with Mousavi et al. (2021) [77], where he showed the following bands responded to EC: 1.910–1.990, 2.102–2.103, 2.109–2.126, 2.138–2.163, 2.365–2.367. In addition, Seifi et al. (2020) [78] pointed out that the spectral reflectance data were positively correlated to soil salinity in a range of 400–1891, 2017 to 2165 and 2280–2359 nm. On the other hand, the spectral reflectance data was found to negatively correlate to soil salinity in a range of 1891–2017, 2166 to 2279 and 2360–2400 nm.

The results showed that the spectral responses of soil calcium carbonate were observed in separate places in the wavelength as follows: 470, 658, 812, 1158, 1440, 1564, 1860 and 2262, respectively, and their correlation coefficient values were −0.185, −0.170, −0.0975, 0.0459, 0.0679, 0.0852, 0.0946, and 0.107, respectively. CaCO₃ significantly influences

the reflectance characteristics of soil and has spectral activity in the NIR spectral region (700–2500 nm). The strongest diagnostic vibrational absorptions are at 2300–2350 nm, and the other three weaker bands occur near 2120–2160 nm, 1997–2000 nm and 1850–1870 nm [79]. The soil spectrum characterizes complex absorption patterns with a large number of predictor variables that are highly collinear, and therefore analyses of diffuse reflectance spectra require the use of multivariate calibrations [59]. Figure 4 shows a few other prominent absorption peaks between 2200–2300 nm and around 2440 nm. According to Clark (1999) [79]. Calcium carbonate tends to increase soil brightness [80] and also exhibits diagnostic features in the infrared wavelength region, with the strongest absorption centered near 2300 nm to 2350 nm [81]. The highest values of the regression coefficients had wavelengths in the NIR spectral range of 2325 nm to 2365 nm with a peak at 2340 nm.

*3.3. Estimation of Soil Parameters Using Different Models*

Three multivariate regression models: PLSR, MARS and Support Vector Regression (SVR), was used for modeling the predictability of soil parameters (pH, EC and $CaCO_3$); based on the laboratory data (observed) and soil spectral data throughout 350 to 2500 nm.

3.3.1. Partial Least Square Regression (PLSR) of Soil Parameters (pH, EC and $CaCO_3$)

The outliers of each calibration and validation dataset were removed, whereas the rest values were used for the modeling process. This data was divided in the internal PLSR process into ten components, whereas each component included the same number of variables. Table 5 shows the number of samples used in modeling for pH, EC and $CaCO_3$ after removing the outlier in calibration and validation processing. In addition, the accuracy assessment for both calibration and validation models for selected soil parameters.

**Table 5.** The predictability assessment of the soil parameters using the PLSR model.

| Soil Parameter | Calibration Data-Set | | | | Validation Data-Set | | | |
|---|---|---|---|---|---|---|---|---|
| | *n* | RMSE | RPD | $R^2$ | *n* | RMSE | RPD | $R^2$ |
| pH | 63 | 0.0721 | 2.254 | 0.68 | 26 | 0.0932 | 1.452 | 0.52 |
| EC (dS/m) | 65 | 0.0856 | 1.461 | 0.61 | 27 | 0.1112 | 1.316 | 0.21 |
| $CaCO_3$ (%) | 66 | 0.0995 | 2.465 | 0.55 | 29 | 0.3519 | 1.881 | 0.41 |

Sixty-three measured pH values and their spectral data were used as data entered of the PLSR model for calibration. The RMSE value of the pH calibration model was 0.0721, while RPD and $R^2$ values were 2.254 and 0.68, respectively. The results are consistent with [21,82,83].

The predictability assessment of the pH validation model was achieved, whereas RMSE, RPD and $R^2$ values were 0.0932, 1.452 and 0.52, respectively.

The results of calibration of EC values and their corresponding spectral variables showed that the regression coefficient $R^2$ value was 0.61 while RPD and RMSE were 1.461 and 0.0856 dS/m. The results show that soil salinity can be predicted using spectral signatures, and this is consistent with the results of Zhou et al., 2022 [84]. The validation PLSR model validation of EC by RPD, RMSE and $R^2$ were 1.316, 0.1112 dS/m and 0.21, respectively.

The results of $CaCO_3$ calibration showed that the $R^2$ value was 0.55, RPD was 2.465, and RMSE was 0.099, these results agree with Alomar et al., 2022 [85], where the results were 0.518. 3.39 for $R^2$ and RMSE, respectively. The validation of $CaCO_3$ predicting was: 3519, 1.88 and 0.41 for RMSE, RPD and $R^2$, respectively, as shown in Figure 5.

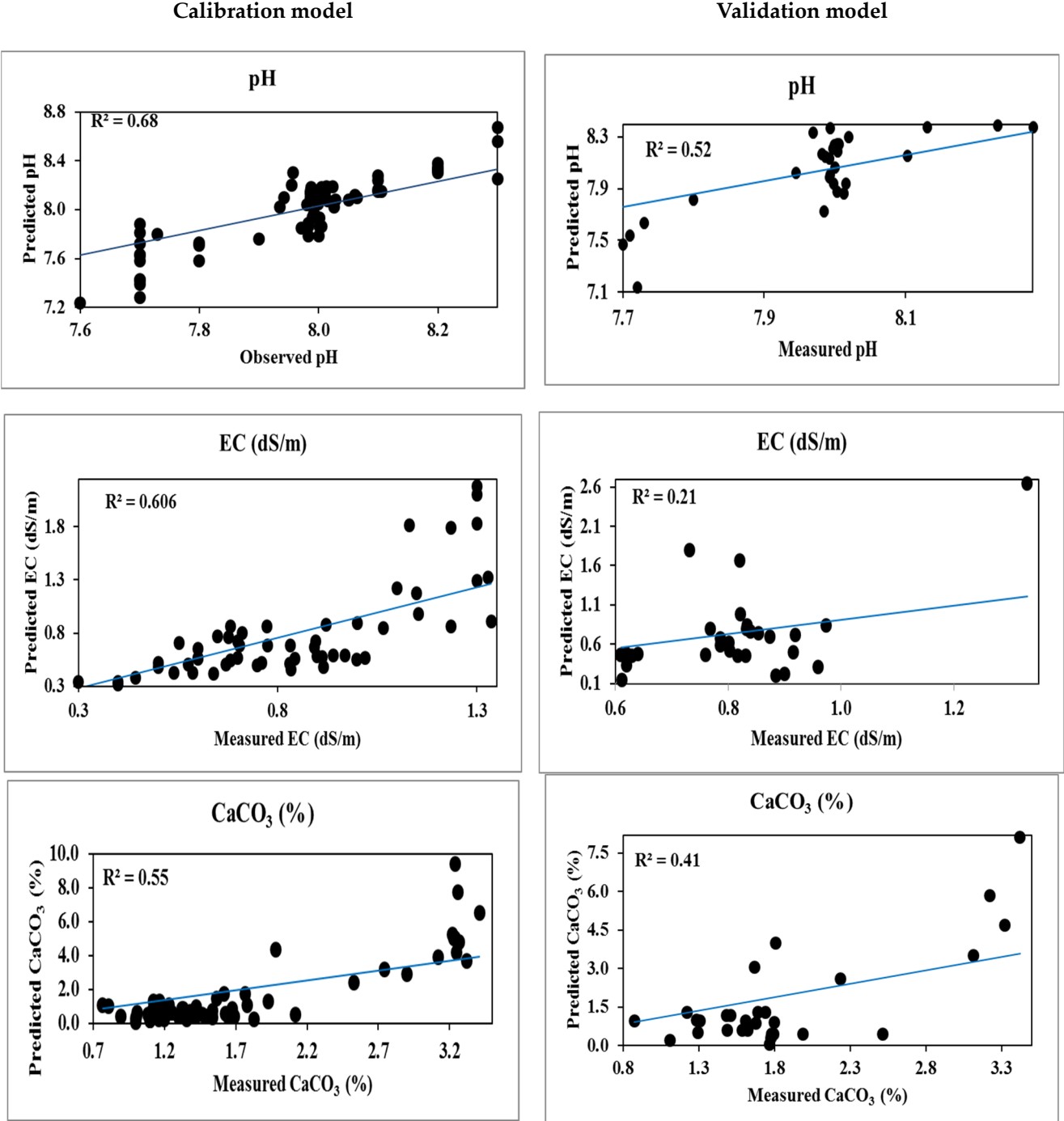

**Figure 5.** PLSR calibration and validation of soil parameters.

For selecting the most correlated or effective bands that were chosen as inputs to derive the PLSR model component for determining the various soil parameters, the CARS technique was applied, as presented in the next Equations (15)– (17).

$$\text{Soil pH parameter} = 0.028 - 0.0258R_{492} + 0.0628R_{828} - 0.0423R_{1276} - 0.0148R_{1158} + 0.0236R_{1636} - 0.01642R_{1656} + 0.0564R_{20681999} + 0.0752R_{2350} \tag{15}$$

$$\text{Soil EC parameter} = 0.032 + 0.0364R_{1014} - 0.0569R_{1194} - 0.0547R_{1222} + 0.0258R_{1276} - 0.0364R_{1410} + 0.02145R_{1516} + 0.0675R_{1602} - 0.0827R_{1626} \tag{16}$$

$$CaCO_3 \text{ parameter} = 0.034 + 0.0358R470 - 0.0837R_{658} + 0.0361R_{812} + 0.0568R_{1158} + 0.0363R_{1440} - 0.02482R_{1564} + 0.0462R_{1860} - 0.0824R_{2262} \tag{17}$$

### 3.3.2. Multivariate Adaptive Regression Splines (MARS)

The process of removing the outliers was performed for the two datasets to enhance the predictability of each soil parameter, as well as to increase the homogeneity of the used data. However, the MARS model was developed for each soil parameter in the investigated area, and the accuracy assessment of each generated model was done and shown in Table 6.

**Table 6.** The obtained data of MARS models for each soil parameter.

| Soil Parameter | Calibration | | | | Validation | | | |
|---|---|---|---|---|---|---|---|---|
| | *n* | RMSE | RPD | $R^2$ | *n* | RMSE | RPD | $R^2$ |
| pH | 66 | 0.125 | 1.737 | 0.59 | 29 | 0.136 | 1.413 | 0.46 |
| EC (dS/m) | 67 | 0.139 | 1.491 | 0.42 | 29 | 0.153 | 0.957 | 0.23 |
| CaCO$_3$ (%) | 67 | 0.256 | 1.421 | 0.58 | 29 | 0.289 | 0.898 | 0.11 |

The obtained data of the MARS models for each soil parameter are explained as follows.

The MARS calibration model performance of the soil pH parameter was tested, whereas the $R^2$ value was 0.59, RMSE was 0.125, and RPD was 1.737. No soil pH values were removed from the validation dataset. The RMSE, RPD, and $R^2$ values for the soil pH validation model were 0.136, 1.413 and 0.46, respectively.

The calibration result estimation of the soil EC was an $R^2$ value of 0.42, while the RPD and RMSE values were 1.491 and 0.139 dS/m, respectively. Moreover, for the MARS validation model, $R^2$, RPD and RMSE were calculated, and their values were 0.23, 0.975, and 0.153 dS/m, respectively.

Furthermore, the results showed that an $R^2$ value of 0.58 was recorded for estimating the soil content of CaCO$_3$, while the $R^2$ value was 0.11 for the validation model. The RMSE value was 0.256 for calibration and validation 0.289 for validation models. The RPD value of the calibration model was 1.421, while 0.898% for the validation, as shown in Figure 6.

The MARS model deals with data as subsets or pieces to detect dependent variables (soil parameter) and a set of predictors (spectral variables) relationships using individual linear regressions with two or more spline functions. These results agree with the results of [86,87], for pH and EC.

### 3.3.3. Support Vector Regression (SVR) Model

The SVR was used for modeling the soil (pH, EC and CaCO$_3$), the outliers were removed, while 65 soil pH values were used for calibration. The results showed that $R^2$, RMSE and RPD values were 0.66, 0.0977 and 1.844, respectively. The SVR validation model (*n* = 28) was able to estimate the soil pH parameter by which $R^2$ = 0.41, RPD = 1.420 and RMSE = 0.113.

To calibrate the SVR model using 66 values of soil EC was conducted. The obtained results of the SVR calibration model showed that $R^2$ of regression was 0.70, while RPD was 1.330, and RMSE was 0.3961 dS/m. In the SVR validation model, lower performance was recorded ($R^2$ = 0.26, RPD = 0.555, and RMSE = 0.369 dS/m) for predicting the soil EC parameter.

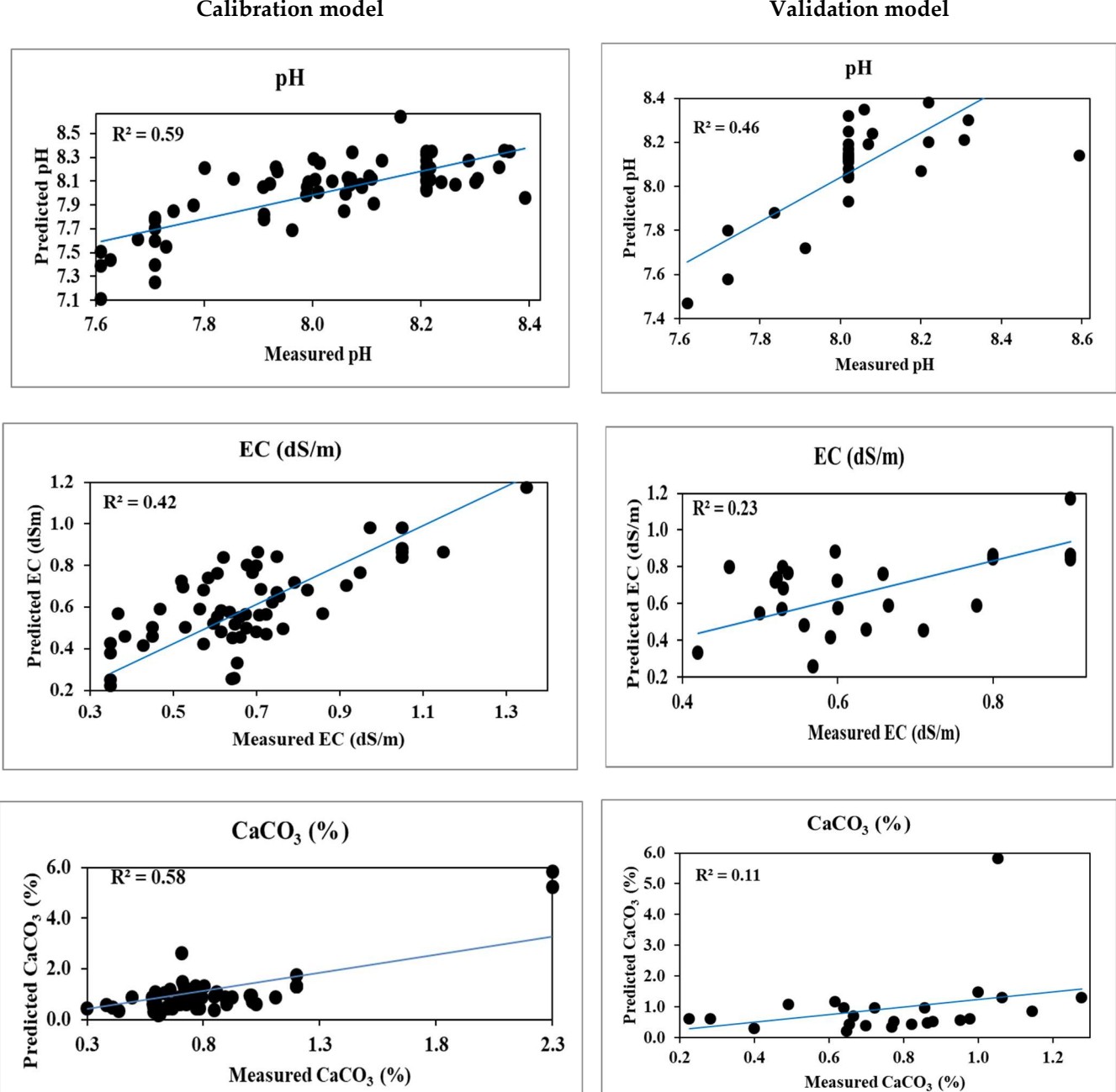

**Figure 6.** MARS calibration and validation of soil parameters.

Two CaCO$_3$ outliers were removed, and the rest data were used for calibration and validation. The R$^2$, RPD and RMSE in the SVR calibration model were 0.74, 1.784, and 0.7953%, respectively, while the validation results in the SVR model were (R$^2$ = 0.47, RPD = 1.247, and RMSE = 0.666%). The obtained results showed that the SVR has the advantage compared with PLSR and MARS for predicting pH, EC and CaCO$_3$. Their other studies support our results, such as [88,89].

The randomization process for selecting the entire data in each dataset is very important in SVR modeling of soil parameters to ensure the un-bias of the obtained outputs of both calibration and validation processes. Mouazen et al. (2010) [90] adopted a similar approach, where the latent variables obtained from PLSR were used as input to a Back Propagation Artificial Neural Network (BPNN), not to SVR as done in the current work.

As PLSR and MARS models, SVR was assessed for its performance in predicting and estimating soil pH, EC and CaCO$_3$ and the values of the evaluation parameters were presented in Table 7.

**Table 7.** The obtained data of SVR models for each soil parameter.

| Soil Parameter | Calibration | | | | Validation | | | |
|---|---|---|---|---|---|---|---|---|
| | *n* | RMSE | RPD | R$^2$ | *n* | RMSE | RPD | R$^2$ |
| pH | 65 | 0.0977 | 1.844 | 0.66 | 28 | 0.113 | 1.420 | 0.41 |
| EC (dS/m) | 66 | 0.3961 | 1.330 | 0.70 | 29 | 0.369 | 0.555 | 0.26 |
| CaCO$_3$ (%) | 66 | 0.7953 | 1.784 | 0.74 | 29 | 0.666 | 1.247 | 0.47 |

The observed and predicted soil parameters data scatter plot based on SVR calibration and validation models are shown in Figure 7.

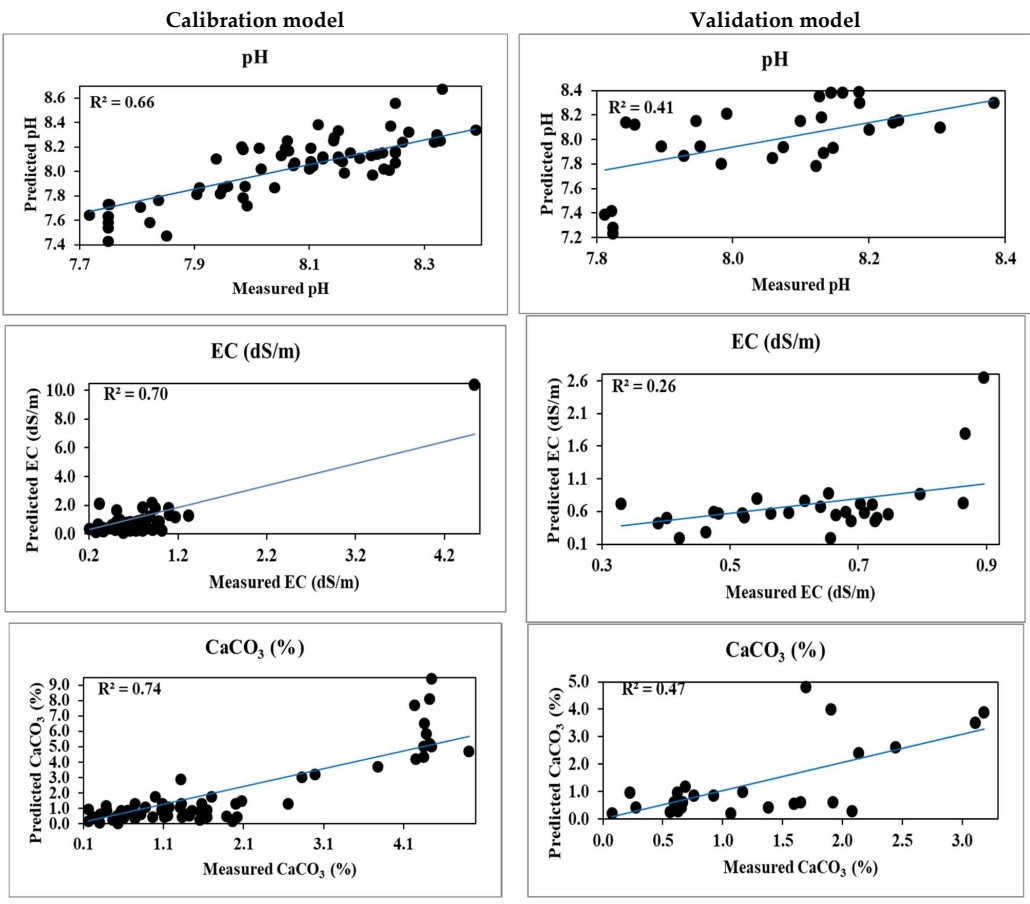

**Figure 7.** SVR calibration and validation of soil parameters.

*3.4. The Machine Learning Models for Predicting Soil Parameters*

The RF and ANN machine learning algorithms were used in this study to associate a large number of inputs and resulting outputs set with higher predictability through three steps of training, testing and validating and predicting the various soil parameters (pH, EC and CaCO$_3$). However, the RF and ANN outputs are discussed in the following parts.

3.4.1. Random Forest (RF)

By the bragging process, training data is developed through a combination of randomly selected variables at each node to mature a tree. The same process occurs in the

following steps of testing and validation. The obtained data of the RF model of calibration and validation are demonstrated in Table 8.

**Table 8.** The obtained data of RF models for each soil parameter.

| Soil Parameter | Calibration | | | | Validation | | | |
|---|---|---|---|---|---|---|---|---|
| | *n* | RMSE | RPD | $R^2$ | *n* | RMSE | RPD | $R^2$ |
| pH | 65 | 0.0572 | 2.975 | 0.82 | 28 | 0.0686 | 2.339 | 0.57 |
| EC (dS/m) | 66 | 0.15800 | 2.231 | 0.78 | 29 | 0.1082 | 2.343 | 0.81 |
| CaCO$_3$ (%) | 67 | 0.3568 | 3.268 | 0.83 | 29 | 0.2978 | 2.659 | 0.75 |

Figure 8 showed the fitting of each RF calibration and validation model, whereas the observed pH values were plotted and the estimated pH values. As calibration and validation accuracy parameters, $R^2$, RPD and RMSE were calculated, $R^2$ = 0.82 for calibration and 0.57 for validation. The RPD of the calibration was 2.975, and the RPD of the validation was 2.339. The RMSE values of the RF calibration and validation models were 0.0572 and 0.0686, respectively.

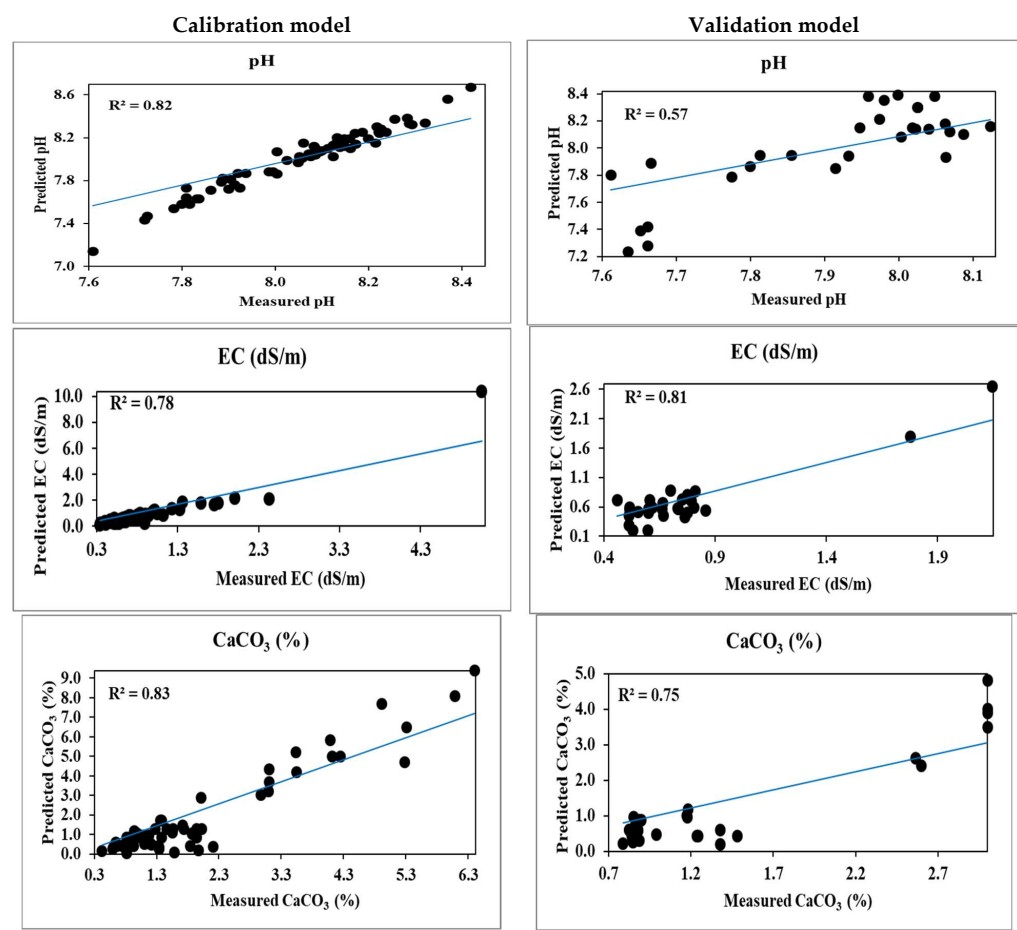

**Figure 8.** RF calibration and validation of soil parameters.

The obtained results of modeling EC using the RF calibration model showed that RMSE, RPD and $R^2$ were 0.1580, 2.231 and 0.78, respectively. Meanwhile, the validation models were 0.1082, 2.343 and 0.81 for RMSE RPD and $R^2$, respectively. This result was matched with [91,92].

The obtained results of modeling CaCO$_3$ using the RF calibration model showed that RMSE, RPD and $R^2$ were 0.3568, 3.268 and 0.83, respectively. Meanwhile, hand the

validation models were 0.2978, 2.659 and 0.75 for RMSE, RPD and $R^2$, respectively. The RF scatter plots, calibration, and validation models are presented in Figure 8.

### 3.4.2. Artificial Neural Network (ANN)

The ANN consists of input, hidden and output layers with connected neurons (nodes) to estimate the unknown inputs. The inputs are the soil combined with observed data from the laboratory and spectra. An ANN model was performed to predict soil pH, EC and $CaCO_3$ and its outputs are presented in Table 9.

**Table 9.** The obtained data of ANN models for each soil parameter.

| Soil Parameter | Calibration | | | | Validation | | | |
|---|---|---|---|---|---|---|---|---|
| | n | RMSE | RPD | $R^2$ | n | RMSE | RPD | $R^2$ |
| pH | 65 | 0.2391 | 1.438 | 0.69 | 15 | 0.2592 | 1.385 | 0.53 |
| EC (dS/m) | 66 | 0.4866 | 2.248 | 0.96 | 15 | 0.5016 | 1.869 | 0.64 |
| $CaCO_3$ (%) | 65 | 1.670 | 1.149 | 0.55 | 14 | 1.723 | 1.114 | 0.53 |

The training model of ANN was tested for its accuracy, whereas RMSE was 0.2391, RPD was 1.438, and $R^2$ was 0.69 for soil pH. Regarding the ANN validation model, $R^2 = 0.53$, RMSE = 0.2592, and RPD = 1.385.

Figure 9 shows the scatter plotting of observed EC values and predicted EC values for training and validating steps using the ANN model.

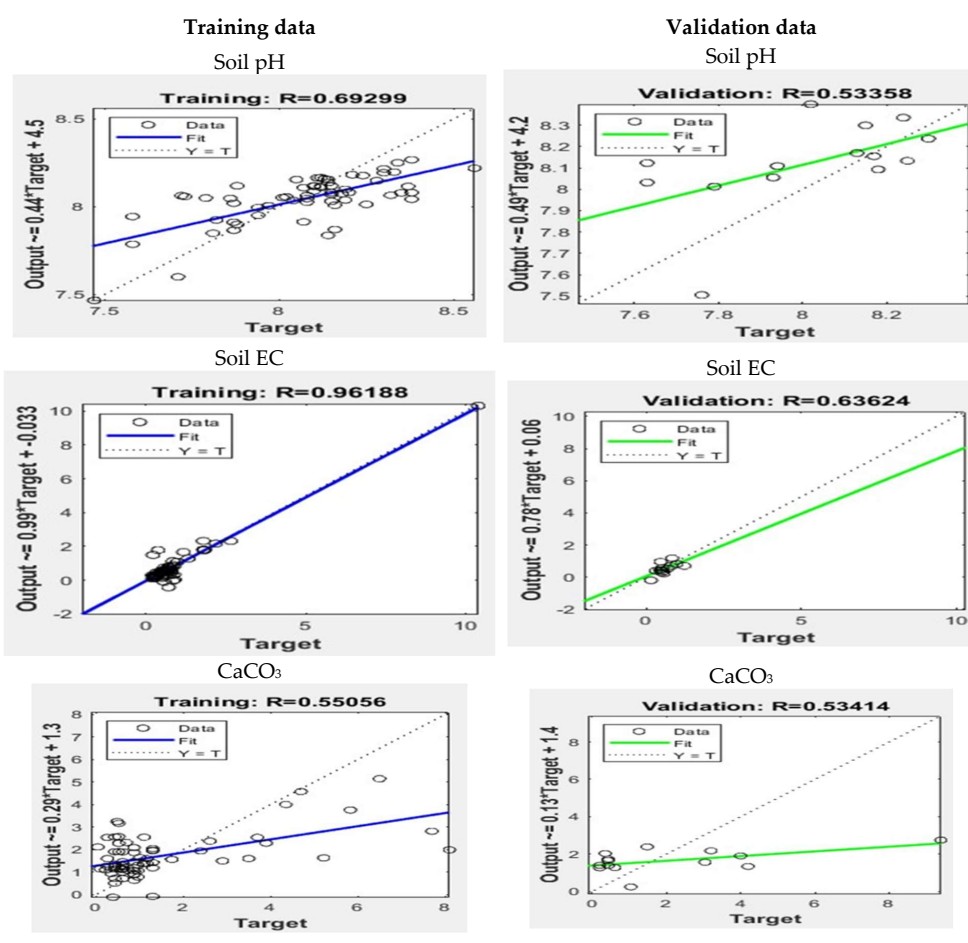

**Figure 9.** ANN training/calibration and validation of soil parameters.

The Calibration model of EC showed high performance as $R^2$ was 0.96, RMSE was 0.486, and RPD was 2.248.

The ANN validation model showed that $R^2$, RPD and RMSE values were 0.64, 1.869, and 0.5016 dS/m, respectively.

The predictability of the soil $CaCO_3$ was evaluated, with $R^2$ as 0.55 in the ANN training step, while it was 0.53 in the validation step. The RPD values in the ANN training and validation were 1.149 and 1.114, respectively, while the RMSE values for the two ANN models were 1.670 and 1.723%.

The relationship between measured and predicted soil parameters was demonstrated in Figure 9 for training and validation ANN models, respectively.

## 4. Conclusions

The current study aims to characterize and develop prediction models and evaluate the accuracy and predictability of Soil EC, pH and $CaCO_3$ in some soils using vis-NIR spectroscopy data in arid lands (the Elkobaneyya valley). Three multivariate regression models, PLSR, MARS, and SVR and two machine learning models were used. The PLSR model is the best model for predicting soil pH in terms of calibration and validation, where $R^2 = 0.68$ and 0.52, and the SVR model was the best model for predicting soil EC and $CaCO_3$, where they were $R^2$ 0.70 and 0.74, respectively for calibration. Meanwhile, the $R^2$ values were 0.26 and 0.47 for validation. The RF and ANN have good results in predicting these parameters. The result showed that the best model to predict soil pH and $CaCO_3$ is RF, where $R^2$ values were 0.82 and 0.83 for calibration, and 0.57 and 0.75 for validation, respectively. Furthermore, the best model for predicting soil EC was ANN; the $R^2$ was 0.96 for calibration and 0.8 for validation for selecting the most correlated bands or effective for predicting the various soil parameters. CARS technique was applied to disengage the high-response bands to soil pH; the significant bands were 492, 828, 1276, 1158, 1636, 1656, 2068, and 2350. Whereas for soil EC parameters, the significant bands were 1014, 1194, 1222, 1276, 1410, 1516, 1602 and 1626. Additionally, regarding the soil $CaCO_3$, the significant bands were 470, 658, 812, 1158, 1440, 1564, 1860 and 2262.

The final results showed that RF has advantages over ANN in predicting the PH and $CaCO_3$ in calibration and validation. Additionally, ANN has the advantage of predicting the EC more than RF.

**Author Contributions:** Conceptualization, M.E.F., A.H.A.-E., A.R.A.M., E.S.M., D.E.K. and M.A.E.-S.; methodology, M.E.F., A.H.A.-E., A.R.A.M., E.S.M., D.E.K. and M.A.E.-S.; software, M.E.F., A.H.A.-E., A.R.A.M., E.S.M., D.E.K. and M.A.E.-S.; validation, M.E.F. and A.R.A.M.; formal analysis, M.E.F., A.H.A.-E., A.R.A.M. and M.A.E.-S.; investigation, M.E.F., A.H.A.-E.; resources, M.E.F., A.H.A.-E., A.R.A.M. and M.A.E.-S.; data curation. M.E.F., A.H.A.-E., A.R.A.M. and M.A.E.-S.; writing original draft preparation, M.E.F., A.H.A.-E., A.R.A.M. and M.A.E.-S.; writing review and editing, M.E.F., A.H.A.-E., A.R.A.M., E.S.M., D.E.K. and M.A.E.-S.; visualization, M.E.F. and A.R.A.M.; supervision, M.E.F., A.H.A.-E., A.R.A.M., E.S.M., D.E.K. and M.A.E.-S.; project administration, M.E.F., A.H.A.-E., A.R.A.M., E.S.M., D.E.K. and M.A.E.-S.; funding acquisition. All authors have read and agreed to the published version of the manuscript.

**Funding:** This research received no external funding.

**Institutional Review Board Statement:** Not applicable.

**Informed Consent Statement:** Not applicable.

**Data Availability Statement:** Not applicable.

**Acknowledgments:** The manuscript presented a scientific collaboration between scientific institutions in two countries (Egypt and Russia). The authors would like to thank the Aswan, Sohag, Al Azhar, RUDN Universities and National Authority for Remote Sensing and Space Science (NARSS) for funding the field survey satellite data and spectral measurements using ASD device. Furthermore, this paper was supported by the RUDN University Strategic Academic Leadership Program.

**Conflicts of Interest:** The authors would like to hereby certify that there are no conflicts of interest in the data collection, analyses, interpretation, the writing of the manuscript or the decision to publish the results. The authors also would like to declare that the funding of the study has been supported by the authors' institutions and universities.

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
