# Peer review of "Integration Vis-NIR Spectroscopy and Artificial Intelligence to Predict Some Soil Parameters in Arid Region: A Case Study of Wadi Elkobaneyya, South Egypt"

_agronomy, doi:10.3390/agronomy13030935_

Round 1
Reviewer 1 Report
In the manuscript titled Integration Vis-NIR spectroscopy and Artificial Intelligence to Predict some Soil Parameters in Arid Region: A Case Study of Wadi Elkobaneyya – South Egypt, El-Sayed et.al used three multivariate regression models: PLSR, MARS and LS-SVR; and two machine learning algorithms:RF and ;ANN to characterize and develop prediction models for soil salinity, pH and CaCO3 by using integration soil analysis and spectral reflectance vis-NIR spectros copy.
On the one hand, I found the paper to be generally well written and for the most part well described. I believe the authors performed a careful and thorough comparison of the experimental results of the model. On the other hand, I found the paper to be overly detailed in some of the descriptions of the models and inadequate or missing altogether in some very important points. I have little confidence in an important analysis, such as the screening of spectral variables, etc. I regret that there is not much innovation in this paper and finally raise some issues that prevent me from recommending this paper for publication. Therefore, I recommend a major revision. I explain my concerns in more detail below. I ask the authors to be specific in their responses about each of my comments.
Major comments:
1. The authors provide detailed information in Methods about Model’s Introduction,which is too voluminous and cumbersome and could be more concise.
2. The question is about the full rationality of the conclusion. In the modelling of organic matter spectra of soils, the pre-processing method of spectral data and the modelling method can affect the prediction results. The screening of spectral variables is an important step in the study of soil VIS-NIR spectra, which can effectively remove redundant information from the spectra and improve the prediction accuracy of the model. The only pre-processing of the spectral information I see in the paper is simply changing the interval of the spectra and nothing else. Further research is needed on how to optimise the spectra and remove the interferences in the soil to build a more robust prediction model.
3 The current manuscript needs to be polished by a native English speaker or a professional language editing service.
Minor comments:
1 Line 34, Forget to add R2 after this sentence:PLSR model was the best model for soil pH parameter.
2 Line 36, The first occurrence of "EC" is recommended for full spelling.
3 Line 41, "R²=64", clearly wrong.
4 Line 570, Missing ")" .
5. The validation "R²=0.8" is wrong, it should be "R²=0.64".
Author Response
Major comments:
- The authors provide detailed information in Methods about Model’s Introduction, which is too voluminous and cumbersome and could be more concise.
We have reviewed the introduction and removed some paragraphs which may reduce as possible.
- The question is about the full rationality of the conclusion. In the modelling of organic matter spectra of soils, the pre-processing method of spectral data and the modelling method can affect the prediction results. The screening of spectral variables is an important step in the study of soil VIS-NIR spectra, which can effectively remove redundant information from the spectra and improve the prediction accuracy of the model. The only pre-processing of the spectral information I see in the paper is simply changing the interval of the spectra and nothing else. Further research is needed on how to optimize the spectra and remove the interferences in the soil to build a more robust prediction model.
Actually we optimized the data spectra in the prepossessing , we clarified this in section 24.2 page 6 , and also we checked the accuracy results using RPD ratio and correlation coefficient (R2) values, category (A) includes soil parameters which are highly predictable with R-square between 0.8 and 1, and with RPD above 2.
The current manuscript needs to be polished by a native English speaker or a professional language editing service.
We have reviewed the English language again.
Minor comments:
All comments have been considered and addresses in this revised version
Reviewer 2 Report
Dear Authors,
I revised the manuscript "Integration Vis-NIR spectroscopy and Artificial Intelligence to Predict some Soil Parameters in Arid Region: A Case Study of Wadi Elkobaneyya – South Egypt" submitted to Agronomy journal. The topic presented by you on forecasting parameters characterizing the soil environment is interesting and necessary for agricultural practice. In my opinion, the manuscript was not fully developed, please pay special attention to enriching the discussion of the results. Please comply with the following comments:
Introduction:
1. Please add a short paragraph in which you characterize soils in Egypt.
2. What physical properties of the soil allow you to use spectral methods to evaluate it? water content? color? Please include in the text.
Material and Methods:
3. Table 1. Please center. I suggest using borders.
4. I suggest you paste some photos showing the excavated soil profiles with a description - at least for the dominant soils in the study area.
5. Why did you put together the described modeling methods? Are you comparing linear methods with nonlinear modeling?
6. Why such proportions in the division between the teaching and validation set? Why did you not separate the test set?
7. Lack of information about the type of neural network chosen for analysis, the method of learning the network, how many neurons were the networks made of?
Results:
8. Which parameter was the most important for you to evaluate the quality of the generated models?
9. Which parameter was most important to you for evaluating the quality of the models' prediction?
10. Line 507. Change background.
11. Figure 9. Please improve the graph - align it with the other graphs in the paper.
12. Refine the discussion of the results - please add at least 15 papers from the last 10 years in which the authors create and use predictive models in agriculture.
13. What maximum values of forecast errors do you think are acceptable in this type of research?
14. What is the advantage of nonlinear models over linear models in the forecasting presented?
References:
15. Add DOI references where possible.
Author Response
- Please add a short paragraph in which you characterize soils in Egypt.
We have considered and added a paragraph in front of the introduction.
- What physical properties of the soil allow you to use spectral methods to evaluate it? water content? color? Please include in the text.
In this paper soil chemical properties (Total calcium carbonate (CaCO3), Soil reaction (pH) and soil salinity (EC)) were predicted.
- Table 1. Please center. I suggest using borders.
We have considered and added the border
- I suggest you paste some photos showing the excavated soil profiles with a description - at least for the dominant soils in the study area.
Unfortunately It is not available now, where we collected surface samples 0-5 cm.
- Why did you put together the described modeling methods? Are you comparing linear methods with nonlinear modeling?
We use several methos linear and nonlinear methods to chick the best model to predict soil properties (pH, EC and CaCO3); based soil spectral data
- Why such proportions in the division between the teaching and validation set? Why did you not separate the test set?
Actually, we randomly split the soil analysis and its spectral reflectance as 70% of for calibration and 30% for validation of the model accuracy.
- Lack of information about the type of neural network chosen for analysis, the method of learning the network, how many neurons were the networks made of?
The Neural Network Approach was completely explained in paragraph 2.4.2.5.
- Which parameter was the most important for you to evaluate the quality of the generated models?
We used R2, RMSE and RPD to chick the model quality , section 2.4.3
- Line 507. Change background.
We changed .
- Figure 9. Please improve the graph - align it with the other graphs in the paper.
We improved the figures .
- Refine the discussion of the results - please add at least 15 papers from the last 10 years in which the authors create and use predictive models in agriculture.
We have considered and supported our results with more 15 updated references in discussion .
- What maximum values of forecast errors do you think are acceptable in this type of research?
Table 3 showed the limitations of acceptable models
- What is the advantage of nonlinear models over linear models in the forecasting presented?
Actually we tried in this manuscript to highlight several method to predicted soil properties . In addition the difference between linear and nonlinear regression models isn’t as straightforward as it sounds. You’d think that linear equations produce straight lines and nonlinear equations model curvature. Unfortunately, that’s not correct. Both types of models can fit curves to data—so that’s not the defining characteristic. In this paper.
- Add DOI references where possible.
We have considered
Reviewer 3 Report
The article requires a thorough read, as there are many grammatical, punctuation errors and missing\unnecessary spaces. Lots of inconsistent sentences, the text is hard to read. The tables are incorrectly designed. After reading the article, there is a feeling of carelessness.
General questions:
1. What was used as the dependent variables for the modelling of soil parameters? Bands? They detail description is necessary.
2. Why talk about 20 profiles (line 20) if there is no further information about them?
3. The summary statistics of soil properties is necessary.
Some notes:
Line 26. Remove comma after “were”
Line 31. Do not start sentence with numbers.
Line 45. Check key words and spaces
Line 132. What is it “Fe”?
The quality if fig.1. is low. Replace please
Line 571 “machine”
Line 577 “values”
...there are many errors...
Although the research is interesting, I suggest, that its presentation is not suitable for publication.
Author Response
- What was used as the dependent variables for the modeling of soil parameters? Bands? They detail description is necessary.
The modeling was dan based soil chemical analysis (pH, EC and CaCO3) and spectral measurements by ASD. we explained in section 2.4 it its subsections
Why talk about 20 profiles (line 20) if there is no further information about them?
This sentence was modified , we used surface soil samples
- The summary statistics of soil properties is necessary.
Table showed statistics on soil properties
- The minor issues
We have modified and considered all comments
Round 2
Reviewer 2 Report
The authors correctly addressed my comments.
Reviewer 3 Report
Corrections have been made